# Moving towards Personalized Medicine in Muscle-Invasive Bladder Cancer: Where Are We Now and Where Are We Going?

**DOI:** 10.3390/ijms21176271

**Published:** 2020-08-29

**Authors:** Juan Carlos Pardo, Vicenç Ruiz de Porras, Andrea Plaja, Cristina Carrato, Olatz Etxaniz, Oscar Buisan, Albert Font

**Affiliations:** 1Department of Medical Oncology, Catalan Institute of Oncology, University Hospital Germans Trias i Pujol, Ctra. Can Ruti-Camí de les Escoles s/n, 08916 Badalona, Spain; jcpardor@iconcologia.net (J.C.P.); aplaja@iconcologia.net (A.P.); oetxaniz@iconcologia.net (O.E.); 2Catalan Institute of Oncology, Badalona Applied Research Group in Oncology (B·ARGO), Ctra. Can Ruti-Camí de les Escoles s/n, 08916 Badalona, Spain; vruiz@igtp.cat; 3Germans Trias i Pujol Research Institute (IGTP), Ctra. Can Ruti-Camí de les Escoles s/n, 08916 Badalona, Spain; 4Pathology Department, Hospital Germans Trias i Pujol, 08916 Badalona, Spain; criscarrato@gmail.com; 5Urology Department, University Hospital Germans Trias i Pujol, 08916 Badalona, Spain; obuisan2903@gmail.com

**Keywords:** muscle-invasive bladder cancer, chemotherapy, immunotherapy, personalized medicine, predictive biomarker

## Abstract

Neoadjuvant cisplatin-based chemotherapy followed by radical cystectomy is the recommended treatment, with the highest level of evidence, for patients with muscle-invasive bladder cancer (MIBC). However, only a minority of patients receive this treatment, mainly due to patient comorbidities, the relatively small survival benefit, and the lack of predictive biomarkers to select those patients most likely to benefit from this multimodal approach. In addition, adjuvant chemotherapy has been recommended for patients with high-risk MIBC, although randomized trials have not provided conclusive evidence on the impact of this approach. At present, however, this situation is changing, largely due to our improved knowledge of the molecular biology of bladder cancer, which has enabled us to identify new prognostic and predictive biomarkers that can be used to select the most appropriate treatment for each patient. Moreover, new active treatments, especially immunotherapy, have shown promising results in the neoadjuvant setting. In addition, the gene expression profile of bladder tumors can be used to classify them into different subtypes, which correlate with specific clinical-pathological characteristics and with treatment response or resistance. Therefore, the main objective for the near future is to introduce these translational breakthroughs into routine clinical practice in order to personalize treatment for each patient.

## 1. Neoadjuvant Therapy in Muscle-Invasive Bladder Cancer (MIBC): An Overview

The prognosis of bladder cancer (BC) has not improved significantly over the last 30 years, as few advances have been made in treatment options since the introduction of cisplatin-based chemotherapy in the mid-1980s. While it is true that platinum-based schemes have somewhat improved survival of patients with advanced disease, less than 50% of patients initially respond and most eventually develop resistance. In platinum-resistant patients, second-line chemotherapy provides poor survival expectations [1]. Despite these limitations, chemotherapy has been incorporated into the treatment of muscle-invasive bladder cancer (MIBC) in combination with local treatments, such as cystectomy or radiotherapy. However, the limited survival benefit and the absence of predictive biomarkers have led to some reluctance to incorporate these strategies into the routine management of the disease, especially with regard to the use of neoadjuvant chemotherapy (NAC) in MIBC. Although international clinical guidelines recommend NAC with cisplatin-based schemes with the highest level of evidence [2,3], less than 20% of patients with MIBC receive this treatment in clinical practice [4]. There are two main reasons for this: firstly, cisplatin-based chemotherapy is contraindicated in approximately 50% of patients due to renal failure or other comorbidities [5]; and secondly, there is a lack of clinical and pathological markers to identify patients who will benefit from NAC, while delaying local treatment in those who will not benefit can have a negative impact on prognosis [6].

Fortunately, this discouraging scenario has changed in recent years, mainly due to two advances. Firstly, the molecular characterization of BC has made it possible to identify potential prognostic and predictive biomarkers, as well as new therapeutic targets. Secondly, the recent introduction of new active therapies, especially immunotherapy, has expanded the therapeutic arsenal in BC beyond platinum-based chemotherapy.

Importantly, it is essential to identify biomarkers to select those patients likely to benefit from these novel treatment options, including immunotherapy—either alone or in combination with chemotherapy and/or biological therapies. This would also help to avoid the administration of ineffective treatments that may lead to unnecessary toxicities, delays in the administration of more effective therapies, and lowered cost-effectiveness. Studies with DNA and RNA sequencing have demonstrated the heterogeneity of bladder tumors at the molecular and genetic level and have identified specific profiles—based on differences in the type and frequency of mutations, in gene copy numbers, and in methylation patterns—that can help define patient prognosis and sensitivity to specific treatments [7]. However, despite these advances, only around 40% of bladder tumors present genomic alterations that are potentially treatable with targeted therapies. Moreover, in contrast to other cancers, such as melanoma, gastrointestinal stromal tumors, and breast and lung cancers, in which 20–60% of patients can be treated with therapies selected by the analysis of biomarkers [8], no biomarkers for BC have been approved for use in clinical practice. It is crucial, therefore, to transfer advances in molecular biology as quickly and efficiently as possible into clinical practice if we are to improve the survival of patients.

Here, we describe the potential applications of molecular biology in the diagnosis and treatment of BC, and especially of MIBC, based on the concept of precision medicine.

## 2. Predictive Biomarkers of Response to Cisplatin-Based Chemotherapy

Cisplatin-based chemotherapy remains the standard treatment for BC both for advanced disease and for the perioperative or conservative treatment of localized BC [2]. The antitumor mechanism of action of cisplatin is based on the formation of adducts that cause DNA damage, which prevents cell replication and induces cell death [9]. Cisplatin induces DNA damage either as single-strand breaks (SSBs), double-strand breaks (DSBs), or interstrand-crosslinks. However, cancer cells have several mechanisms to repair DNA damage. SSBs can be repaired by base excision repair, mismatch repair, or nucleotide excision repair (NER), while DSBs can be repaired by non-homologous end joining or homologous recombination (HR). The analysis of genes involved in the different pathways of DNA damage repair (DDR) may enable us to identify predictive biomarkers of response to cisplatin [10].

The NER pathway, which repairs SSBs, includes the ERCC excision repair 1 (ERCC1) and ERCC excision repair 2 (ERCC2) proteins. High levels of ERCC1 indicate a gain of NER pathway function, leading to greater repair of the DNA damage caused by cisplatin and thus decreased efficacy of the drug [11]. Bellmunt et al. demonstrated that in patients with metastatic BC treated with a cisplatin-based combination, those with high *ERCC1* expression had worse prognosis [12]. Several studies have analyzed the role of *ERCC2* mutations as predictive biomarkers in BC. *ERCC2* mutations are found in 12% of BCs, mostly in the helicase domain of the gene, and have been associated with a loss of NER pathway function in preclinical studies [13]. Interestingly, *ERCC2* mutations confer sensitivity to both cisplatin and carboplatin, but not to treatments with doxorubicin, ionizing radiation or poly (ADP-ribose) polymerase (PARP) inhibitors [13]. Whole exome sequencing of 50 MIBC tumor samples treated with cisplatin-based NAC demonstrated that *ERCC2* mutations were significantly associated with treatment response. Specifically, *ERCC2* mutations were detected in 36% of patients who responded to chemotherapy (<ypT1), while no mutations were detected in non-responders (>ypT2) [14]. These results were confirmed in a subsequent validation study [15]. In the joint analysis of the two series, *ERCC2* mutations were found in 38% (17/45) of responders and in only 6% (3/53) of non-responders [13]. Taken together, these data suggest that ERCC2 may well be a predictive biomarker of response to cisplatin-based chemotherapy. In this line, a recent study by the Memorial Sloan Kettering Cancer Center (MSKCC) compared the genomic profile of primary vs. secondary MIBC patients treated with NAC. They found that not only was the benefit of NAC limited to primary MIBC but also that *ERCC2* mutations were observed more frequently in primary MIBC (12% vs. 1.2%), which could explain the lack of benefit of NAC in secondary MIBC [13]. Other studies have also correlated mutations in the ATM serine/threonine kinase (*ATM*), RB transcriptional corepressor 1 (*RB1*) and FA complementation group C (*FANCC*) repair genes with efficacy of NAC in MIBC. In the study of Plimack et al., 13 of 15 cisplatin-responders (87%) presented mutations in some of these genes, while none of the non-responders harbored these mutations. In their validation study, 64% of responders presented some of these mutations as compared to only 15% of non-responders [16]. In a recent update of this study, a significant improvement in the five-year disease-specific survival was observed in carriers of at least one mutation compared to patients without mutations (90% vs. 49%, *p* = 0.0015) [17]. The presence of mutations in DDR pathways can identify patients likely to respond to NAC, who could be potential candidates for a bladder-sparing approach. The ongoing phase II RETAIN trial in patients treated with NAC aims to identify candidates for bladder preservation among patients who attain a complete response and have mutations in *ATM*, *RB1*, *FANCC,* or *ERCC2* (NCT02710734).

HR is used to repair DSBs. In a molecular profile analysis using the NGS600 testing platform, mutations in HR genes were detected in 17% of 17,566 tumors and in 23% of BCs, which had the third highest frequency of HR-DDR mutations (after endometrial and biliary tract carcinomas). The most frequently mutated genes in BC were AT-rich interaction domain 1A (*ARID1A*) (12.4%), *ATM* (4%), BRCA1 DNA repair associated (*BRCA1*) (3%), and BRCA2 DNA repair associated (*BRCA2*) (4.5%) [18]. More recently, 13% of patients with BC were found to harbor germline variants, 75% of which were located in DDR genes, mainly in *BRCA2*, mutS homolog 2 (*MSH2*), *BRCA1*, checkpoint kinase 2 (*CHEK2*), ERCC excision repair 3 (*ERCC3*), nibrin (*NBN*), and RAD50 double strand break repair protein (*RAD50*) [19]. In this line, a study by our group in 57 MIBC patients treated with NAC, increased *BRCA1* mRNA expression negatively correlated with pathological response and survival [20].

DDR is a complex process involving several different DNA repair pathways, making it necessary to analyze extensive panels of genes if we are to determine the predictive and prognostic value of the genes involved in each pathway. An MSKCC study analyzed a panel of 34 DDR genes in 100 patients with advanced BC treated with platinum-based chemotherapy and detected at least one alteration in one of the genes in 47 patients. Median overall survival was significantly higher in these patients than in those with no alterations (23.7 vs. 13.0 months, *p* = 0.006). Interestingly, the survival benefit was observed in both cisplatin- and carboplatin-treated patients [21]. A recent phase II trial in 49 patients treated with neoadjuvant dose-dense cisplatin plus gemcitabine analyzed a panel of 29 DDR genes and found that the presence of deleterious mutations, including *ERCC2* mutations, was associated with response to chemotherapy, with a positive predictive value of 89% and a two-year relapse-free survival of 100% [22].

In summary, current evidence indicates that the analysis of alterations in DNA repair pathways can provide prognostic and predictive information in BC patients. However, prospective studies including a larger number of patients are required to confirm these findings. In addition, given that cisplatin is contraindicated in approximately 50% of patients, it is important to analyze these biomarkers in patients treated with carboplatin as well. Finally, the study of these alterations will pave the way for the discovery of new prognostic and predictive biomarkers as well as the incorporation of new biological therapies, such as PARP inhibitors, which have been shown to be effective in ovarian and breast cancer patients harboring mutations in HR genes [23].

## 3. Therapeutic Implications of BC Molecular Subtypes

Several research groups have proposed various classifications of BC molecular subtypes based on the two reference subtypes, luminal and basal (Table 1). A group from Lund University analyzed a large number of non-MIBC and MIBC tumors and proposed five subtypes: urobasal A, urobasal B, genomically unstable, infiltrated, and squamous-like [24]. Urobasal A and B are characterized by the expression of biomarkers usually expressed in the normal urothelium, while the squamous-like subtype expresses keratin 5 (KRT5), keratin 6 (KRT6), and keratin 14 (KRT14), which are specific to squamous differentiation. The infiltrated subtype is characterized by stromal and immune cell infiltration. A group from the University of North Carolina (UNC) proposed another classification in which the basal subtype presents sarcomatoid characteristics and expresses high levels of both epidermal growth factor receptor (EGFR) and its ligands, while the luminal subtype expresses epithelial markers (E-cadherin (CDH1) and miR-200) and alterations in fibroblast growth factor receptor 3 (FGFR3) [25]. An MD Anderson (MDA) group added a third subtype, called p53-like, which is characterized by the presence of stromal markers and the activation of tumor protein p53 (TP53) [26]. The Cancer Genome Atlas (TCGA) proposed a classification based on four molecular subtypes called clusters (I–IV) [7]. Cluster I corresponds to the luminal phenotype and presents characteristics of papillary tumors, cluster II has the luminal phenotype but with a predominance of p53-like characteristics, and clusters III and IV correspond mainly to the basal subtype defined in the UNC and MD Anderson classifications [25,26]. A recent update of the TCGA study [27] defined five molecular subtypes: luminal papillary (35%), luminal infiltrated (19%), luminal (6%), basal-squamous (35%), and neuronal (5%). The TCGA luminal subtypes are characterized by a high expression of urothelial differentiation markers, such as forkhead box A1 (*FOXA1*), GATA binding protein 3 (*GATA3*), and peroxisome proliferator activated receptor gamma (*PPARG*). The luminal-papillary subtype has *FGFR3* mutations, amplifications, overexpression, and FGFR3-TACC3 fusions. The luminal-infiltrated subtype, which corresponds to cluster II in the original TCGA classification [7], is characterized by elevated expression of epithelial-mesenchymal transition (EMT) markers, such as twist family bHLH transcription factor 1 (*TWIST1*) and zinc finger E-box-binding homeobox 1 (*ZEB1*), and moderate expression of the immune markers Programmed death ligand 1 (*PDL1*), and cytotoxic T-lymphocyte associated protein 4 (*CTLA4*). The luminal subtype presents high expression of the keratin 20 (*KRT 20*). The basal-squamous subtype is defined by the expression of CD44 antigen (*CD44*), *KRT5*, *KRT6*, *KRT14*, is enriched in *TP53* mutations, and has the highest expression of the immune markers *PD-L1*, Programmed cell death 1 (*PD-1*) and *CTLA4*. Finally, the neuronal subtype has high expression of neuroendocrine and neuronal markers [27]. A Canadian study proposed four subtypes: basal, luminal, luminal-infiltrated, and claudin-low [28]. The claudin-low subtype corresponds to TCGA cluster IV [7] and has characteristics of the basal subtype, with the expression of EMT markers and immune infiltration. Recently, the Bladder Cancer Molecular Taxonomy Group (BCMTG) has proposed a consensus classification based on the analysis of 1750 transcriptomic profiles of the classifications published to date [29]: papillary luminal (24%), unspecified luminal (8%), unstable luminal (15%), stromal rich (15%), basal-SCC (35%), and neuroendocrine-like (3%). A web application of this model allows individual and anonymous classification of tumor samples according to this consensus (http://cit.ligue-cancer.net:3838/apps/consensusMIBC_web/).

The molecular subtypes have been associated with specific clinical-pathological characteristics and differential sensitivity to treatments. Basal-SCC tumors, identified in the TCGA, UNC, and MD Anderson classifications [7,25,26], predominate in women and are associated with more aggressive tumors, more advanced disease stages, worse prognosis, and squamous cell features. In contrast, luminal tumors seem to be less aggressive but more resistant to NAC [27,28]. Recently, Lotan et al. showed that in early-stage MIBC treated with cystectomy, upstaging (≥pT3) is less likely in luminal than non-luminal tumors [30], suggesting that luminal tumors could be managed more conservatively with upfront cystectomy.

These molecular classifications can complement or help reconsider the standard histological classifications, since squamous differentiation might well be underreported. For example, only 42% of BCMTG [29] basal-SCC tumors had squamous cell features in the histological analysis. The histological, genomic, and transcriptional heterogeneity of BC has important clinical implications. Warrick et al. analyzed the intratumoral heterogeneity of different regions of primary MIBC tumors in relation to the Lund molecular subtypes [24] and histological variants. Nearly 40% of the tumors demonstrated molecular heterogeneity among the different histologies, especially the basal-squamous tumors, 78% of which co-occurred with either urothelial-like or genomically unstable tumors [31]. These results emphasize the need for an adequate tissue sampling that selects different areas of the tumor when establishing the molecular subtype.

Several studies have indicated that it is possible to classify BC into molecular subtypes using immunohistochemistry (IHC), which is less complex than transcriptomic analysis and could facilitate the use of molecular subtypes in clinical practice. Markers related to basal subtypes, such as KRT5/6, KRT14 and p63, and those associated with luminal phenotypes, such as GATA3, FOXA1, uroplakin and erb-b2 receptor tyrosine kinase 2 (HER2), have been proposed for an IHC-based classification. A meta-analysis found that by analyzing only GATA3 and KRT5/6 with IHC, it was possible to identify the basal and luminal subtypes with 91% reliability [32]. Recently, Makboul et al. stratified BC patients according to the Lund classification [24] using a simple IHC panel of five biomarkers (FGFR3, CK5, cyclin-B1, HER2, p53). More than 90% of tumors were classified without overlap and the different tumor subtypes significantly correlated with prognosis [33]. The different subtypes have also been correlated with response to chemotherapy. Basal tumors are associated with a better response to cisplatin-based chemotherapy, while tumors classified as p53-like or those in cluster II [7] have been associated with chemoresistance [26]. An MD Anderson study of 60 MIBC patients treated with neoadjuvant dose-dense methotrexate + vinblastine + doxorubicin + cisplatin (M-VAC) plus bevacizumab found that patients with basal tumors had more pathological complete responses (pCR) and longer survival than those with luminal or p53-like tumors. Five-year survival rates for patients with basal, luminal, and p53-like tumors were 91%, 73%, and 36%, respectively, probably due to the greater sensitivity to NAC in basal tumors, since achieving downstaging is a predictive factor for longer survival. Additionally, only patients with p53-like tumors presented bone metastases at disease progression. Interestingly, this study also found a greater frequency of the p53-like subtype in cystectomy samples than in transurethral resection (TUR) samples, especially in luminal tumors, suggesting that NAC can induce switching of tumor subtypes [34]. The Canadian study [28] analyzed the association between their four tumor subtypes (basal, luminal-infiltrated, luminal, claudin-low) and response to NAC. Patients with luminal tumors had the longest overall survival, while those with claudin-low tumors had the shortest, regardless of whether they received NAC or only cystectomy. In contrast, patients with basal tumors had longer survival but did not fare differently from luminal tumors if they were treated with NAC rather than only cystectomy. Recently, the same group has defined four subtypes by transcriptional analysis in residual tumor at cystectomy after NAC: CC1-basal, CC2-luminal, CC3-immune, and CC4-scar-like. The basal and luminal phenotypes observed in the residual disease were similar to the pretreatment subtypes. The CC3-immune tumors had the worst outcome and showed a high immune infiltration, suggesting a potential positive impact for immune checkpoint inhibitors (ICIs) as second-line treatment, whereas the CC4-scar-like tumors showed a low proliferation rate, expressed fibrosis, and had a good prognosis regardless of response to NAC [35]. These findings suggest that establishing molecular subtypes after NAC in residual tumor disease can be useful in selecting adjuvant treatment.

A recent study by our group, using IHC-based hierarchical clustering, classified MIBC patients treated with NAC in three clusters: BASQ-like (FOXA1/GATA3 low; KRT5/6/14 high), luminal-like (FOXA1/GATA3 high; KRT5/6/14 low), and mixed-cluster (FOXA1/GATA3 high; KRT5/6 high; KRT14 low). Patients with BASQ-like tumors were more likely to achieve a pathological response to NAC (OR 3.96; *p* = 0.017) [36].

Taken together, these findings indicate that the molecular classification of BC according to gene expression profiles can play an important role in selecting the most effective treatment for each patient. Thus, basal tumors would benefit most from chemotherapy, while luminal tumors would be associated with better prognosis but poor response to cisplatin-based chemotherapy, indicating that the best treatment option for these patients could be cystectomy. Molecular classification can thus provide additional information to the standard histological classification and better characterize BC for personalized treatment approaches.

In recent years, with the emergence of immunotherapy as a treatment option for BC, several studies have attempted to establish a correlation between tumor subtypes and immunotherapy efficacy. The revised TCGA classification [27] suggested that patients with luminal-infiltrated tumors and especially those with basal tumors can derive the greatest benefit from immunotherapy. Importantly, however, the TCGA patients had localized tumors and had not received any previous treatment, thus, these treatment strategies require validation in future studies. The IMvigor 210 study [37] demonstrated a greater benefit of treatment with atezolizumab, a PD-L1-blocking antibody, in advanced BC classified as TCGA cluster II [7], and in the CheckMate 275 study, basal tumors responded better to nivolumab, a PD-1-blocking antibody [38]. Surprisingly, tumors with high immune infiltration, classified as claudin-low, showed a poor response to immunotherapy [37,38]. This apparent paradox could be explained by the fact that there is a more effective suppression of T cells in cluster IV than in cluster II tumors [39]. Intriguingly, immunotherapy and chemotherapy seem to be effective in complementary patient populations. Patients with luminal-infiltrated tumors (cluster II) would benefit from immunotherapy, while in those with basal tumors (cluster IV), chemotherapy may be the treatment of choice. Although patients with neuronal-subtype tumors generally have a worse prognosis, a recent analysis of the IMvigor 210 trial showed that TCGA neuronal-subtype tumors [27] responded better to atezolizumab [40]. Moreover, this did not seem to be associated with other parameters related to immunotherapy response; for instance, the tumor mutational burden (TMB) and the load of tumor neo-antigens were lower in these tumors than in the other subtypes and none of the tumors were immunoinflammatory. In contrast, the neuronal subtype had low levels of transforming growth factor beta 1 (TGF-β1), which has been associated with improved response to immunotherapy [41]. In summary, while a molecular classification can help to select the best treatment option for each patient, it will be necessary to take into consideration other factors that may affect response to either chemotherapy or immunotherapy.

## 4. Predictive Biomarkers of Immunotherapy Response

The emergence of immunotherapy has highlighted the importance of biomarkers when deciding on the optimal treatment for each patient. In addition to molecular subtypes, many potential biomarkers have been correlated with the response to immunotherapy: PD-L1 expression, CD8^+^ T-cell infiltration, DDR gene alterations, TMB, and immune and stromal gene expression signatures such as the interferon gamma (IFN-γ) signature [42]. To date, unfortunately, none of these markers has shown sufficiently consistent results to warrant incorporation into the routine management of BC.

PD-L1 expression is detected in 20–30% of bladder tumors and is associated with more advanced disease and worse prognosis [43]. Studies of advanced BC have shown conflicting results regarding the role of PD-L1 as a predictive biomarker of response to immunotherapy [44]. Importantly, since PD-L1 is a dynamic biomarker both in space and time, the analysis of a small tumor fragment in a biopsy may not be representative of the PD-L1 expression in the whole tumor. Moreover, prior treatment may influence PD-L1 expression. Therefore, IHC results of PD-L1 expression must be interpreted in the context of broader biomarker panels when selecting patients to receive immunotherapy.

Some biomarkers, such as the neutrophil/lymphocyte ratio, albumin levels, high C-reactive protein, and Interleukin-6 levels, can be easily incorporated into routine clinical practice, while others, such as gene expression signatures, are more complex [45]. In addition to the potential usefulness of the molecular subtypes in predicting response, the study of DDR pathways may be helpful, since defects in DDR have been associated with an increase in the TMB, and thus, with a greater immune response. An MSKCC study analyzed a panel of 34 DDR genes in patients with advanced BC treated with atezolizumab or nivolumab and found a significant benefit for immunotherapy in patients with deleterious mutations in the genes [46]. In addition, ICIs have recently been shown to be highly effective in tumors with defects in the MMR/microsatellite instability pathway [47]. In an effort to encompass these different biomarkers that may be related to immunotherapy response, an immunogram has recently been proposed that incorporates in seven main axes the different parameters related to immunotherapy response, which will help to predict the efficacy of ICIs in individual patients [48].

The possibility of incorporating immunotherapy in earlier stages of BC, where other treatments are currently available, makes it essential to identify biomarkers to select the most effective therapy for each patient. The solid rationale for exploring the efficacy of immunotherapy in early-stage disease has recently been elegantly reviewed [49]. Early stages have a greater integrity of the immune system and can induce greater T-cell expansion than advanced stages, where increased impairment of T-cell function is more evident and where cancer-associated inflammation has been linked to poor response to immunotherapy. Moreover, the neoadjuvant setting is optimal for exploring the role of potential predictive biomarkers to immunotherapy since tumor tissue can be obtained just before treatment initiation and the genomic profile can be compared before and after therapy.

Two recent phase II trials have explored the role of ICIs in the neoadjuvant setting. In the PURE-01 trial [50], 50 patients, most of whom were eligible for cisplatin therapy, were treated with three cycles of pembrolizumab, a PD-1-blocking antibody, followed by cystectomy. In the ABACUS trial [51], 95 patients who were ineligible for cisplatin-based NAC received two cycles of atezolizumab before cystectomy. A pCR was attained by 42% and 31% of patients in the PURE-01 and ABACUS trials, respectively. Both trials included detailed biomarker analyses to define potential predictive biomarkers of response to ICIs. In the PURE-01 study, a significant association between pCR and PD-L1 expression, TMB, and DDR and *RB1* gene alterations was observed. In the ABACUS trial, in contrast, these biomarkers did not correlate with pathological response; however, the quality of immune infiltration measured by CD8 and granzyme B (GZMB), a surrogate marker of activated CD8 cells, as well as an eight-gene cytotoxic T-cell transcriptional signature, significantly correlated with pCR. In addition, the inflamed and desert immune phenotypes, as described by Mariathasan et al. [41], correlated with response and resistance to atezolizumab, respectively. Interestingly, CD8 levels were higher in responding tumors, while high levels of fibroblast activation protein alpha (FAP), a surrogate marker of cancer-associated fibroblasts related to TGF-β, was associated with resistance to immunotherapy. PD-L1, CD8, GZMB, and FAP expression increased post-therapy. These results suggest a potential predictive role of response to ICIs for these markers although the contradictory findings of the two trials indicate a need for validation in a larger number of patients.

Both the PURE-01 and ABACUS trials showed promising results that suggest a level of efficacy for neoadjuvant immunotherapy comparable to that of NAC, making it a feasible treatment option for cisplatin-ineligible patients. This possibility raises the question of how to select cisplatin-eligible patients for NAC or immunotherapy or NAC-plus-immunotherapy, especially considering that many biomarkers, such as TMB and DDR alterations, are associated with the efficacy of both treatments [21,46]. A study exploring the association between the tumor microenvironment and outcome in MIBC patients found that higher T-cell inflamed and IFN-γ signature scores were associated with improved outcome in patients treated with bladder-sparing trimodality therapy (TMT) but with worse outcome in those treated with NAC-plus-RC, while high stromal infiltration was associated with poor prognosis in patients receiving NAC-plus-RC but not in those treated with TMT [52]. Along the same lines, immune signatures predicted response to neoadjuvant immunotherapy in patients included in the PURE-01 trial but not in patients treated with NAC [53]. An ongoing phase III trial (NCT03732677) exploring the combination of NAC and immunotherapy will shed further light on this issue.

A related question is how to integrate and sequence the different therapeutic options in the multimodality management of MIBC. Intriguingly, the cytotoxic effect of NAC can generate an immune effect through the activation of CD8^+^ effector T cells and decreasing T_regs_ [54]. The concurrent administration of NAC and immunotherapy could thus hinder the T-cell response if T cells are killed by NAC. This phenomenon may partly explain the limited benefit obtained with NAC-plus-immunotherapy compared to NAC alone in advanced BC [55]. In contrast, a sequential administration of NAC followed by immunotherapy could be a more effective approach. In the TONIC trial in breast cancer, an upregulation of immune-related genes was detected after cisplatin-based chemotherapy, suggesting that NAC may induce a tumor microenvironment more conducive to immunotherapy response [56]. Finally, evidence suggests that neoadjuvant is more effective than adjuvant treatment, based on the greater tumor antigen exposition before tumor resection [49]. The recent results of the IMvigor 010 phase III trial showed that adjuvant atezolizumab did not demonstrate a significant benefit in high-risk MIBC patients treated with cystectomy [57].

## 5. Conclusions

In recent years, we have greatly broadened our understanding of the molecular biology of BC, which has allowed us to identify new prognostic and predictive biomarkers. We also have at our disposal novel therapeutic options, such as immunotherapy, which can improve patient outcome and quality of life. These new effective drugs show promising results but also highlight the question of how to select the optimal treatment for each patient. There is still no biomarker approved for clinical practice and it is crucial to incorporate into clinical practice all the advances in the field of molecular biology as efficiently and rapidly as possible. Only in this way, will we be able to achieve precision medicine and select the most effective treatment for each individual patient.

## Figures and Tables

**Table 1 ijms-21-06271-t001:** Summary of the main characteristics of molecular subtypes of bladder cancer according to different molecular classifications.

Molecular Classification	Patients (*n*)	Subtypes	Histological and Molecular Characteristics	Ref.
Lund University	308 BC	Urobasal A	High expression of *FGFR3*, *CCND1*, *TP63*, and *KRT5*	[24]
		Urobasal B		
		Genomically unstable (GU)	Frequent *TP53* mutations. *CCNE* and *ERBB2* expression and low cytokeratin expression	
		Squamous cell carcinoma-like (SCC)	High expression of basal keratins normally not expressed in the urothelium	
		Infiltrated	Stromal and immune cell infiltration	
UNC	262 High grade MIBC	Luminal	Expression of epithelial markers (E-cadherin/CDH1 and miR-200) and alterations in *FGFR3*	[25]
		Basal	Sarcomatoid features. High expression of EGFR and its ligands	
MDA	73 MIBC	Luminal	Features of active PPARγ and estrogen receptor transcription. *FGFR3* mutations	[26]
		Basal	p63 activation and squamous differentiation	
		p53-like	Presence of stromal markers and activation of p53 signature	
TCGA (2014)	131 High grade MIBC	Cluster I	Luminal phenotype; presence of papillary tumors features	[7]
		Cluster II	Tumors with luminal phenotype but with a predominance of p53-like subtype features	
		Cluster III	Correspond to basal subtype defined in the UNC and MD Anderson classifications	
		Cluster IV		
TCGA (2017)	412 T2-4, N0-3, M0-1 MIBC	Luminal papillary (35%)	*FOXA1*, *GATA3*, and *PPARG* expression. FGFR3 alterations	[27]
		Luminal infiltrate (19%)	*FOXA1*, *GATA3*, and *PPARG* expression. Expression of EMT (high) and immune (moderate) markers	
		Luminal (6%)	*FOXA1*, *GATA3*, and *PPARG* expression. High expression of *KRT 20*	
		Basal-SCC (35%)	High expression of immune response markers. *CD44*, *KRT5*, *KRT6* and *KRT14* expression. *TP53* mutations	
		Neuronal (5%)	High expression of neuroendocrine and neuronal markers	
BCMTG	1750 MIBC transcriptomic profiles	Luminal papillary (24%)	Papillary morphology. Expression of *FGFR3* and *PPARG*. Mutations in *FGFR3* and *KDM6A*	[29]
		Luminal non-specified (8%)	Micropapillary morphology. *PPARG* expression. Mutations in *ELF3*	
		Luminal unstable (15%)	Expression of *PPARG*, *E2F3*, and *ERBB2*. Genomic instability. Mutation in *TP53* and *ERCC2*	
		Stroma-rich (15%)	Stromal and immune cell (B cells) infiltration	
		Basal/squamous (35%)	*EGFR* expression. Mutations in *TP53* and *RB1*. Stromal and immune cell (CD8 T and NK cells) infiltration	
		Neuroendocrine-like (3%)	Neuroendocrine differentiation. Loss of TP53 and RB1. Mutations in *TP53* and *RB1*	

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
