# Peer review of "Moving towards Personalized Medicine in Muscle-Invasive Bladder Cancer: Where Are We Now and Where Are We Going?"

_ijms, 2020, doi:10.3390/ijms21176271_

Round 1
Reviewer 1 Report
P 3, LL 116-120. Please provide the correct citation for the described study detecting mutations in different HR genes (ref 18 does not apply).
P3 LL 130-134. Please provide the number of patients for ref 21.
P12 L 491 citation should be "Eur Urol 2020" (not 2019 as suggested by acceptance date of the paper).
P4 L 179 reconsider link (corrupted?); original link provided in the paper: http://cit.ligue-cancer.net:3838/apps/consensusMIBC_web/
P4 L 188 suggestion for amendment: "can complement or help reconsider the standard histological classifications". Squamous differentiation might be underreported.
P 5 L 201: Ref 31 is not "recent" compared to Refs 26-28
Reviewer 2 Report
Juan Carlos Pedro at al. make a decent effort to summarize and present the advances in MIBC. However, there were parts of the draft that were not very easy to completely follow, especially in the part "Therapeutic implications of BC molecular subtypes". For example, luminal or basal tumors respond better to cisplatin and have the longest survival? It seems that different studies show different results but it should be presented in a more clear way.
Additionally, authors should very briefly explain what are the monoclonal antibodies or cocktail drugs that other studies use as tehrapy for Bladder cancer. For example authors should mention that atezolizumab is a PDL1 antibody, whereas nivolumab and pembrolizumab are PD-1 antibodies and what MVAC stands for (which drugs are included).
Lastly, authors should label Table 1.
